# Target sample mining with modified activation residual network for speaker verification

**Ji Chaoqun** *, **Chen Wei, Ye Peng, Wang Zhou, Zhou Shuhang**

Wenzhou Business College, Wenzhou, Zhejiang, The People's Republic of China

* jicq2022@163.com

**Data availability statement:** The data underlying the results presented in the study are available from https://www.robots.ox.ac.uk/~vgg/data/voxceleb/.

**Funding:** The author(s) received no specific funding for this work.

**Competing interests:** The authors have declared that no competing interests exist.

## Abstract

In the domain of speaker verification, Softmax can be used as a backend for multi-classification, but traditional Softmax methods have some limitations that limit performance. During the training phase, Softmax is used for multi-class training, while the speaker verification stage is a binary classification validation, leading to a discrepancy between the multi-class training in the training phase and the binary classification validation in the verification stage. It is also important to notice the issue of the disparity in the number of positive and negative samples in the sampling process of a binary classification problem. The difference in positive and negative samples can lead to the dominance of negative sample gradients during machine learning training, which can affect the performance of the speaker verification system. During the process of calculating similarity between positive and negative samples, there may be encountered an issue of overlapping similarity scores. If the overlapping portion is too large, it can reduce the discriminability between positive and negative samples, affecting the speaker system's ability to distinguish between positive and negative samples. Considering the relatively compact distribution of positive and negative sample spaces, it is beneficial for enhancing the performance of the speaker system, and focusing more on the learning of difficult samples is conducive to improving the network's convergence and generalization. Thus, this paper introduces an adaptive target function capable of solving these issues (SphereSpeaker). SphereSpeaker introduces different types of hyperparameters on the basis of Softmax, making it more suitable for handling speaker verification problems. SphereSpeaker also introduces three different angular margins to update the network, further enhancing the stability and generalization ability of the network model. Meanwhile, considering the issues of gradient vanishing, gradient explosion, and model degradation that can occur in deep neural networks, this paper introduces a deep neural network, which is named as Residual Network PReLu(ResNet-P). The experimental results indicate that compared to other deep neural network methods, this method has the lowest equal error rate, significantly improving the performance of the speaker verification system.

## Introduction

With the rapid development of deep learning, deep neural networks have achieved good performance in many fields. In the speaker verification domain, the Deep Vector (D-Vector) method [1] has pushed the application of deep learning theories to a climax. It inputs the speaker's voice frames into a Deep Neural Network (DNN) and obtains frame-level embedding features. The X-Vector method [2] extracts speaker features through Time-Delay Neural Networks (TDNN) [3]. Due to the network's time delay characteristics, it can learn more frame-level correlations during the learning process, thereby increasing the robustness of the recognition system. With the performance improvement brought by the X-Vector method, more network architectures have been applied in speaker verification tasks. The Visual Geometry Group-Middle (VGG-M) network [4], initially applied in the field of image processing, consists of 5 convolutional layers and 3 fully connected layers. Due to its outstanding performance in image processing, it has attracted attention from various fields and has been applied in the feature extraction phase of speaker verification tasks [5]. Deep Residual Networks (ResNet) [6] can directly pass shallow data to deep networks, which is beneficial for gradient optimization and speeds up the training efficiency of the network.

The objective function is crucial for the establishment of deep learning models. The most common objective function initially aimed at classification [7]. This type of objective function mainly focuses on research from two perspectives based on Softmax loss [8]. One is to enhance its discriminative ability by increasing the distance between decision boundaries of different classes, including its variants such as Additive Margin Softmax (AM-Softmax) loss [9], Dynamic Additive Margin Softmax [10], and Additive Angular Margin Softmax (AAM-Softmax) loss [11], among others. The second method is to enhance the discriminability of the Softmax loss through regularization, which typically establishes a connection between the regularizer and the Softmax loss in a weighted form. The regularizers used are usually loss functions that can be used independently, such as Center loss [12], Ring loss [13], etc. Common objective functions aimed at metric learning include binary cross-entropy loss [14], contrastive loss [15], triplet loss [16], quadruplet loss [17], and objective functions based on mutual information Adaptive Estimation(MIAD) [18], among others. With the research and development of sampling techniques, methods that optimize solely for metric learning can also achieve ideal performance, with effects similar to those of combining classification with metric learning [19].

## Objective function based on sample mining

The main research is for a target function of sample mining, which is an improvement based on the Softmax loss function targeting multi-classification. Research has found that the Softmax method has some restrictive drawbacks [20]. The Softmax function is used within a multi-class framework during training, whereas the speaker verification phase is a classic binary classification problem, which can lead to discrepancies between the training phase and the verification phase. The multi-classification method during the training process increases the difficulty of data collection due to the need to determine the category of each sample. For the aforementioned issues, this section introduces a binary classification objective function. This method can effectively bridge the gap between the training phase and the validation phase, enhancing the system's performance.

## Binary classification objective function

To address the differences between the training phase and the validation phase, K binary classification tasks were constructed, with K being the number of speakers in the training set. In

the ith binary classification, positive samples use the ith audio sample, while negative samples use other audio samples. A basic binary classification objective function is shown as follows:

$$L_1 = log\left(1 + e^{-W_i^T x - b_y}\right) + \sum_{i \neq y} log\left(1 + e^{W_i^T x + b_i}\right) \qquad (1)$$

Where: $W_i$ is the weight of the i-th binary classifier, x is the feature of the speaker, $y$ is the corresponding label, and $b_i$ is the bias. To transform the problem into an unconstrained space for classification, normalize the binary classifier weights $W_i$ and speaker features x. The transformed objective function is shown as follows:

$$L_2 = log\left(1 + e^{-\cos(\theta_y)}\right) + \sum_{i \neq y} log\left(1 + e^{\cos(\theta_i)}\right) \qquad (2)$$

which: $\theta_i$ is the angle between the weights of the i-th binary classifier $W_i$ and the speaker features x. The speaker confirms that the task is an open set, while the bias $b_i$ in the equation (1) formula pertains to closed set learning, thus the bias $b_i$ is usually removed. The purpose of this binary classification objective function is to minimize $\theta_i$, thereby reducing the loss. However, there are certain issues with binary classification targets during the process of speaker verification tasks. The following chapters will analyze these problems one by one.

## Balancing positive and negative samples problem

In the previous section, we defined K as the number of speakers in the training set. Multi-speaker methods such as triplet loss, contrastive loss, and Softmax loss all address the issue of positive and negative sample balance. Balancing positive and negative samples is to address the issue of highly imbalanced gradients caused by the uneven quantity of positive and negative samples. An imbalanced number of positive and negative samples can have a negative impact on the network.

During the speaker verification process, each batch selects K speakers, resulting in only 1 positive sample, while the number of negative samples is K-1. This disparity in quantity causes the gradient of the negative samples to dominate, reducing the performance of the model. Consider introducing a weighting factor $\lambda$ to balance the gradients of positive and negative samples, with the following updated expression:

$$L_3 = \lambda log\left(1 + e^{-\cos(\theta_y)}\right) + (1 - \lambda)\sum_{i \neq y} log\left(1 + e^{\cos(\theta_i)}\right) \qquad (3)$$

which: $\lambda \in [0,1]$ is a hyperparameter used to balance the gradients of positive and negative samples. $\lambda$ Can be defined according to the number of K speakers selected for each batch, which can be represented as $\lambda = K - 1/K$.

## Difficulty sample mining problem

Difficulty sample mining has always been one of the core issues in the research of deep learning. In the speaker verification task, hard samples are those with lower scores within pairs of samples from the same speaker and those with higher scores within pairs of samples from different speakers. Similarly, easy samples are those with higher scores within pairs of samples from the same speaker and lower scores within pairs of samples from different speakers. The mining of easy and hard samples has a positive impact on neural networks; the loss of hard samples in the objective function is higher than that of easy samples. Therefore, it is

considered to optimize the objective function to make the model pay more attention to hard samples, which is beneficial for improving the network's convergence and generalization.

Study the hard and easy sample mining methods for Softmax loss, define the normalized Softmax [21] loss as $L_n = -log\left(e^{s \cdot \cos(\theta_y)} / \sum_i e^{s \cdot \cos(\theta_i)}\right)$, where s is an adjustable parameter, label $i \neq y$. Fixed $\cos(\theta_i) = 0.2$, defined $\cos(\theta_y) < 0.2$ as hard samples, $\cos(\theta_y) > 0.2$ as easy samples $\cos(\theta_y) \in [-1, 1]$, it can be known through numerical calculations that as the adjustable parameter s increases, the loss of hard samples is higher and more sensitive compared to easy samples. Introducing the parameter s enables the neural network to focus on optimizing difficult samples, enhancing the convergence and generalization of the neural network. Apply the aforementioned method to the objective function proposed in this chapter, introducing a hyperparameter r to adjust the focus of the objective function on difficult samples. The improved objective function is represented as:

$$L_4 = \frac{\lambda}{r} \cdot log\left(1 + e^{-r \cdot \cos(\theta_y)}\right) + \frac{1-\lambda}{r} \cdot \sum_{i \neq y} log\left(1 + e^{r \cdot \cos(\theta_i)}\right) \qquad (4)$$

which: the larger the hyperparameter r, the higher the attention given to difficult samples. Set the parameters $\lambda$ in equation (4) to 1, define $\cos(\theta_y) > 0$ as easy samples, $\cos(\theta_y) < 0$ as hard samples, adjust $\cos(\theta_y) \in [-1, 1]$ as well as the hyperparameter r. Through numerical calculations, it can be seen that as the hyperparameter r increases, the loss of easy samples approaches zero, while the loss of hard samples remains basically unchanged.

## Margin adjustment issue

In the speaker's confirmation task, the number of similar samples in each batch is much smaller than the number of dissimilar samples, thus the space occupied by dissimilar samples is greater than that of similar samples. In the training process, it is necessary to allocate more space for heterogeneous samples, which is beneficial for the stability and flexibility of the model. Therefore, the margin m is introduced, and the objective function is as follows:

$$L_5 = \frac{\lambda}{r} \cdot log\left(1 + e^{-r \cdot (\cos(\theta_y) - m) - b}\right) + \frac{1-\lambda}{r} \cdot \sum_{i \neq y} log\left(1 + e^{r \cdot (\cos(\theta_i) + m) + b}\right) \qquad (5)$$

Wherein: b is a learnable parameter of deviation, the purpose of which is to improve the stability of training. After the improvement, the positive sample boundary is changed to $r \cdot (\cos(\theta_y - m) - b) = 0$, and the negative sample boundary is changed to $r \cdot (\cos(\theta_i) + m) + b = 0$.

## Similarity adjustment

During the training process, it was discovered that there is a difference in the distribution of cosine similarities between positive and negative sample pairs. Among them, the negative samples are more concentrated in the cosine similarity distribution, while the positive samples are more dispersed in the similarity distribution. The difference in the distribution of similarity can lead to an overlap between positive and negative sample pairs in terms of similarity scores, which can affect the distinction between positive and negative sample pairs, and is detrimental to training. Considering the aforementioned issues, a monotonically decreasing function $g(z)$ is introduced to replace the cosine function. Its expression is as follows:

$$g(z) = 2\left(\frac{z+1}{2}\right)^t - 1 \qquad (6)$$

Wherein: $z = \cos(\theta)$, $z \in [-1,1]$. t is a parameter that can control the overlap of positive and negative sample pairs, as t increases, the overlap of positive and negative sample pairs decreases. It is noteworthy that when $t = 1$, $g(z) = \cos(\theta)$. The modified similarity calculation method still has its value range within $[-1,1]$, but it has solved the problem of overlapping positive and negative sample pairs, increasing the discriminability between positive and negative samples, which facilitates model learning. The objective function after the final modification can be represented as:

$$L = \frac{\lambda}{r} \cdot log\left(1 + e^{-r\cdot\left(g\left(\cos(\theta_y)\right)-m\right)-b}\right) + \frac{1-\lambda}{r} \cdot \sum_{i \neq y} log\left(1 + e^{r\cdot\left(g\left(\cos(\theta_i)\right)\right)+m+b}\right) \tag{7}$$

### Adaptive objective function with different angular margins

Added margin adjustment for homogeneous and heterogeneous samples to increase the stability of training. We will compare the margins of other methods, and the final objective function is shown as follows:

$$L_C = \frac{\lambda}{r} \cdot log\left(1 + e^{-r\cdot\left(g\left(\cos(\theta_y)\right)-m\right)-b}\right) + \frac{1-\lambda}{r} \cdot \sum_{i \neq y} log\left(1 + e^{r\cdot\left(g\left(\cos(\theta_i)\right)\right)+m+b}\right) \tag{8}$$

In equation (8), the use of learnable parameter b to adjust the margin for similar and dissimilar samples can be considered as additive margins. The same approach can also incorporate another type of additive margin [9] and integrate gradient separation methods, making the training process relatively stable. The objective function represented by this method can be expressed as:

$$L_A = \frac{\lambda}{r} \cdot log\left(1 + e^{-r\cdot g\left(\cos(\theta_y)\right)-r\cdot Detach\left(g\left(\cos\left(min(\pi,\theta_y+m)\right)\right)-g\left(\cos(\theta_y)\right)\right)-b}\right)$$
$$+ \frac{1-\lambda}{r} \cdot \sum_{i \neq y} log\left(1 + e^{r\cdot g\left(\cos(\theta_i)\right)+b}\right) \tag{9}$$

Among which: $Detach(")$ is the gradient separation function, which allows some network parameters to not participate in the network parameter update, reducing the impact of the branch network on the main network. Similarly, multiplication margins can also be introduced into the objective function [22,23], and the objective function with multiplication margins can be represented as:

$$L_M = \frac{\lambda}{r} \cdot log\left(1 + e^{-r\cdot g\left(\cos(\theta_y)\right)-r\cdot Detach\left(g\left(\cos\left(min\left(m,\frac{\pi}{\theta_y}\right)\cdot\theta_y\right)\right)-g\left(\cos(\theta_y)\right)\right)-b}\right)$$
$$+ \frac{1-\lambda}{r} \cdot \sum_{i \neq y} log\left(1 + e^{r\cdot g\left(\cos(\theta_i)\right)+b}\right) \tag{10}$$

### ResNet-P

A distinctive feature of residual networks is that they have many "residual units", which can be represented as follows:

$$y_1 = h(x_1) + F(x_1, w_1) \tag{11}$$

Where: $x_1$ is the input speaker features of the l -th residual unit, $w_1 = \{ w_{1k} | 1 \leq k \leq K \}$ is the parameters of the l -th residual unit, K is the number of layers of the residual unit, and $F('')$ is the residual function. $h('')$ can be considered as a direct mapping, that is: $h(x_1) = x_1$. Define the $l + 1$ -th input of the residual unit at layer as the speaker feature $x_{l+1}$, which can be represented as:

$$x_{l+1} = f(y_1) \tag{12}$$

Which: The original method $f('')$ used the activation function Rectified Linear Unit(ReLU) [24], the purpose of ReLU is to increase the non-linear connections between neurons in the neural network. This chapter introduces the Parametric Rectified Linear Unit (PReLU) [25] on this basis. PReLU adds a learnable parameter to ReLU, allowing the activation function to adaptively learn according to the true state of the features, which is beneficial for enhancing the network's ability to represent features and accelerating the convergence rate of the network. The calculation method of PReLU is as follows:

$$f(y_1) = \left\{ \begin{array}{ll} y_1, & y_1 > 0 \\ a_1 y_1, & y_1 \leq 0 \end{array} \right. \tag{13}$$

Where: $y_1$ represents the residual unit at layer l, and $a_1$ is a learnable parameter that controls the slope of the negative part. $a_1$ uses momentum update, the expression is as follows:

$$\Delta a_1 := \mu \Delta a_1 + \eta \frac{\partial \sigma}{\partial a_1} \tag{14}$$

Wherein: $\mu$ set momentum to 0. 9, $\eta$ set the learning rate to 0. 005, and $\sigma$ is the target function, $a_1$ with an initial value of 0. 25. It is worth noting that when the learnable parameters $a_1 = 0$, the above equation becomes ReLU. Draw the PReLU and ReLU curves as shown in Fig 1. The coefficients $a_1$ in the third quadrant of the PReLU coordinate graph are not constant; they are learnable parameters. For ease of observation, it is assumed to be a constant in the figure.

Assuming $f('')$ is an identity mapping, that is, equation (12) can be represented as $x_{l+1} = y_1$, substituting it into equation (11) yields:

$$x_{l+1} = x_1 + F(x_1, w_1) \tag{15}$$

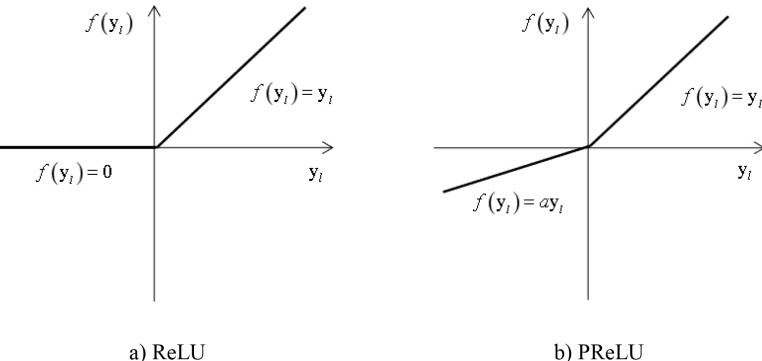

a) ReLU                                    b) PReLU

**Fig 1. Comparison between ReLU and PReLU.**

Obtained through recursive operations in the end:

$$x_{\mathrm{L}} = x_{\mathrm{l}} + \sum_{i=l}^{L-1} \mathrm{F}(x_i, \mathrm{w}_i) \tag{16}$$

Where: $x_{\mathrm{L}}$ is for relatively deep speaker features, $x_{\mathrm{l}}$ is for relatively shallow speaker features, and $\sum_{i=l}^{L-1} \mathrm{F}(x_i, \mathrm{w}_i)$ is for the stacked residual unit. From the above equation, it can be deduced that any deep feature can be represented as a sum of any shallow feature and stacked residual units. We name the residual network with a trainable activation function as the ResNet-P network. This article applies the ResNet-P network to the field of speaker verification.

## Experimental analysis

**Dataset.** The experiment utilized the large-scale speaker recognition database VoxCeleb1 [5] with varying speech quality. The audio in the database is extracted from YouTube videos, which come from various complex environments and contain various types of noise. The development set of the database contains 148,642 speech audio segments provided by 1,211 speakers (690 males, 561 females). The evaluation set includes 40 speakers outside of the development set categories, totaling 4874 speech samples. During testing, the official test plan list is used, with a total of 37720 tests, and the ratio of non-target to target tests is 1:1.

**Procedure.** Process the data next. Since there may be silent segments in the data, it is necessary to first detect the speech and remove the silent parts. Next, feature extraction is performed, with the front-end features using MFCC features with a dimension of 13. Then perform preprocessing operations such as pre-emphasis, framing, and windowing on the voice signal. The pre-emphasis coefficient is set to 0.97, the window length for windowing is 25ms, the frame shift is 10ms, and the number of points for the FFT is set to 512. After the above operations, a spectral feature of the $512 \times 300$ word can be obtained. The residual network that introduces PReLU is denoted as ResNet-P. The last fully connected layer of ResNet-P has a dimension of 256, and the corresponding embedding feature dimension is also 256-dimensional. The optimization algorithm of ResNet-P uses the Stochastic Gradient Descent (SGD) algorithm, with an initial learning rate of $10^{-2}$ and a final learning rate of $10^{-3}$.

## Performance comparison and analysis

Compare the performance of the ResNet-P network with the three proposed objective functions under different parameters. The three methods mentioned above all use Cosine Distance Scoring (CDS) for speaker matching, abbreviated as ResNet-P+SphereSpeaker-C, ResNet-P+SphereSpeaker-A, and ResNet-P+SphereSpeaker-M, respectively. The comparative methods include the speaker's confirmation of traditional statistical models versus deep neural network models. Among them, statistical model-based methods include GMM-UBM, I-vector + PLDA. he front-end acoustic features of GMM-UBM are respectively based on Mel-frequency cepstral coefficients (MFCC) features [26], modified power normalized cepstral coefficients (MPNCC) features [27], and features based on affine transformation and feature transformation (ATFS) [27]. The deep neural network models include four speaker recognition systems, each using ResNet34 [6] as the network architecture and targeting different objective functions: contrastive loss, triplet loss, AM-Softmax loss, and MIAD loss [18], abbreviated as ResNet34+Contrastive, ResNet34+Triplet, ResNet34+AM-Softmax, and ResNet34+MIAD, respectively. In addition, the comparative methods also include: CNN-based methods (AutoSpeech) [28], VGG-based networks [29], SincNet networks [30], SimCLR+NN

[31], BPCSR [32], BGLCC [33]. The performance evaluation criteria use the Equal Error Rate (EER) and the minimum Detection Cost Function (minDCF), with the parameters for minDCF set to the official values. The lower the values of EER and minDCF, the better the performance. Based on the experimental setup above, the performance comparison of different methods is shown in Table 1.

As can be seen from Table 1:

(1) The ResNet-P+SphereSpeaker-A method performs optimally in terms of parameters $\lambda = 0.8, t = 3$ compared to other methods. The equal error rate (EER) reached a minimum value of 6. 17, and the minimum detection cost function (minDCF) reached a minimum value of 0. 48.

(2) Under different objective functions, as $t$ increases, both EER and minDCF decrease when the parameters $\lambda$ are the same. Prove that the parameter $t$ enhances the neural network's ability to distinguish between the similarity of positive and negative samples.

**Table 1. Performance comparison under different $\lambda$, t.**

| Method | $\lambda$ | t | EER(%) | minDCF |
|---|---|---|---|---|
| MFCC+GMM-UBM[26] | – | – | 15. 00 | 0. 80 |
| MPNCC+GMM-UBM[27] | – | – | 8. 05 | 0. 86 |
| ATFS+GMM-UBM[27] | – | – | 7. 23 | 0. 76 |
| ResNet34+MIAD[18] | – | – | 6. 44 | 0. 60 |
| AutoSpeech(N=8,C=128)[28] | – | – | 8. 95 | – |
| VGG[29] | – | – | 7. 00 | 0. 68 |
| SincNet[30] | – | – | 7. 20 | – |
| SimCLR+NN[31] | – | – | 8. 55 | – |
| BPCSR[32] | – | – | 7. 29 | 0. 60 |
| BGLCC[33] | – | – | 10. 07 | – |
| ResNet34+Contrastive[18] | – | – | 7. 98 | 0. 69 |
| ResNet34+Triplet[18] | – | – | 7. 87 | 0. 68 |
| ResNet34+AM-Softmax[18] | – | – | 7. 34 | 0. 62 |
| ResNet-P+SphereSpeaker-C | 0. 6 | 2 | 7. 74 | 0. 54 |
| | 0. 6 | 3 | 6. 47 | 0. 52 |
| | 0. 7 | 2 | 6. 95 | 0. 53 |
| | 0. 7 | 3 | **6. 17** | **0. 52** |
| | 0. 8 | 2 | 8. 23 | 0. 59 |
| | 0. 8 | 3 | 7. 99 | 0. 58 |
| ResNet-P+SphereSpeaker-A | 0. 6 | 2 | 6. 98 | 0. 54 |
| | 0. 6 | 3 | 6. 19 | 0. 50 |
| | 0. 7 | 2 | 7. 08 | 0. 53 |
| | 0. 7 | 3 | 6. 30 | 0. 52 |
| | 0. 8 | 2 | 7. 60 | 0. 57 |
| | 0. 8 | 3 | **6. 17** | **0. 48** |
| ResNet-P+SphereSpeaker-M | 0. 6 | 2 | 7. 61 | 0. 57 |
| | 0. 6 | 3 | 6. 86 | 0. 53 |
| | 0. 7 | 2 | 6. 98 | 0. 54 |
| | 0. 7 | 3 | 6. 93 | 0. 53 |
| | 0. 8 | 2 | 7. 64 | 0. 57 |
| | 0. 8 | 3 | **6. 61** | **0. 52** |

## Parameter selection and analysis

Compare the performance of the ResNet-P network with the three proposed objective functions under different parameters. All three objective functions have two adjustable parameters $\lambda, t$, $\lambda$ is the positive and negative sample equilibrium parameters, $t$ is the similarity adjustment of positive and negative samples. The experimental parameters in this section were controlled by the control variable method, and other parameters except parameters $\lambda$ and $t$ were fixed, to verify the performance differences of the system under different parameters $\lambda$ and $t$. The selection range of the setting parameter is $\lambda \in \{0.6, 0.7, 0.8\}$, $t \in \{2, 3\}$. Use EER and minDCF as performance evaluation metrics. By arranging and combining two parameters, the experimental results are plotted as shown in Fig 2.

As can be seen from Fig 2:

(1) Keep the control parameters $\lambda$ unchanged, adjust the value of parameter $t$ and find that, the values of EER and minDCF for the three methods at $t = 3$ are reduced to varying degrees compared to the values of EER and minDCF at $t = 2$, wherein the decline is relatively significant when $\lambda = 0.8$, the ResNet-P+SphereSpeaker-A method reduces EER by 1.43% and minDCF by 0.09, while the ResNet-P+SphereSpeaker-M method decreases EER by 1.03% and minDCF by 0.05. Prove that the parameter t in the proposed objective function plays a role in distinguishing the overlapping parts of positive and negative sample similarities under different angular margins, thereby improving the performance of the speaker system.

(2) When keeping the control parameter t constant, it can be visually seen that the methods ResNet-P+SphereSpeaker-C and ResNet-P+SphereSpeaker-A have a decrease in EER and minDCF values when $\lambda \in \{0.6, 0.7\}$. As the parameter $\lambda$ decreases, the minDCF value corresponding to the method ResNet-P+SphereSpeaker-M gradually decreases when $t = 3$, $\lambda \in \{0.6, 0.7, 0.8\}$. It can demonstrate that the parameters $\lambda$ of the three objective functions can effectively address the imbalance between positive and negative samples during the training process, enhancing the accuracy of the speaker verification task.

## Convergence comparison and analysis

Conduct a comparative convergence analysis of the three methods proposed in this paper: ResNet-P+SphereSpeaker-C, ResNet-P+SphereSpeaker-A, and ResNet-P+SphereSpeaker-M, with the experimental data settings being the same as in the previous sections. Convergence curves use EER and minDCF as performance evaluation metrics, and all experiments in this paper have an iteration count of 45. Draw convergence curves for the three methods under different parameters, as shown in Fig 3.

As can be seen from Fig 3:

(1) As the number of epoch increases, the error rate (EER) and minDCF for all three proposed methods show a decreasing trend under different parameters. Among them, ResNet-P+SphereSpeaker-C and ResNet-P+SphereSpeaker-A have lower EER, and ResNet-P+SphereSpeaker-A has the lowest minDCF value.

(2) The method proposed in this paper, ResNet-P+SphereSpeaker-A, achieves the lowest minDCF value of 0.48 when the parameters $\lambda = 0.8$, $t = 3$. Further demonstrate the positive promoting effect of the method proposed in this paper on the speaker's task.

(3) The method mentioned in this paper, ResNet-P+SphereSpeaker-C, achieves the lowest EER value of 6.17% when the parameters $\lambda = 0.7$, $t = 3$. ResNet-P+SphereSpeaker-A achieves the lowest EER value of 6.17% when the parameter $\lambda = 0.8$, $t = 3$. Compared to

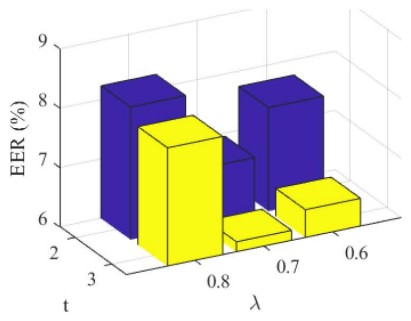

a) ResNet-P+SphereSpeaker-C Method EER

Diagram

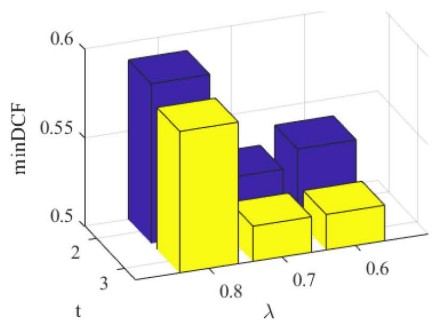

b) ResNet-P+SphereSpeaker-C Method

minDCF Diagram

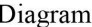

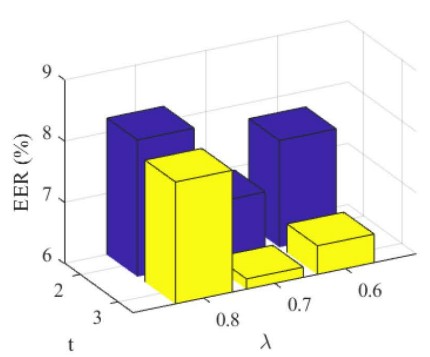

c) ResNet-P+SphereSpeaker-A Method EER

Diagram

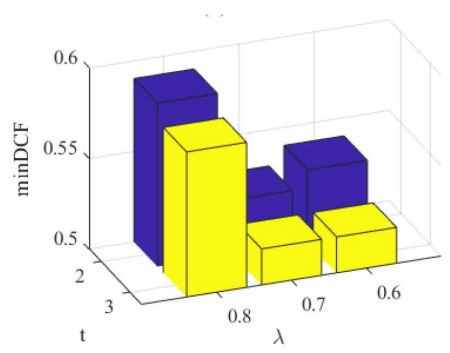

d) ResNet-P+SphereSpeaker-A Method

minDCF Diagram

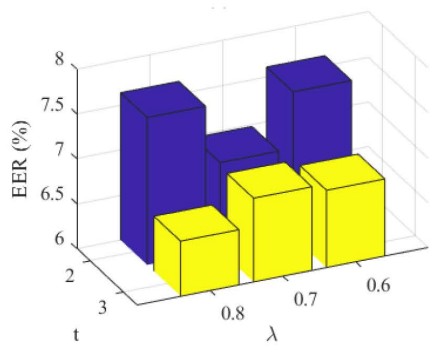

e) ResNet-P+SphereSpeaker-M Method EER

Diagram

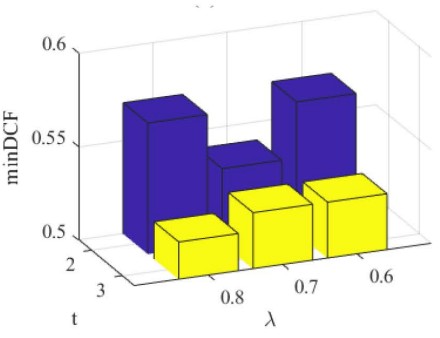

f) ResNet-P+SphereSpeaker-M Method

minDCF Diagram

**Fig 2. Performance of different parameters of the method proposed in this paper.**

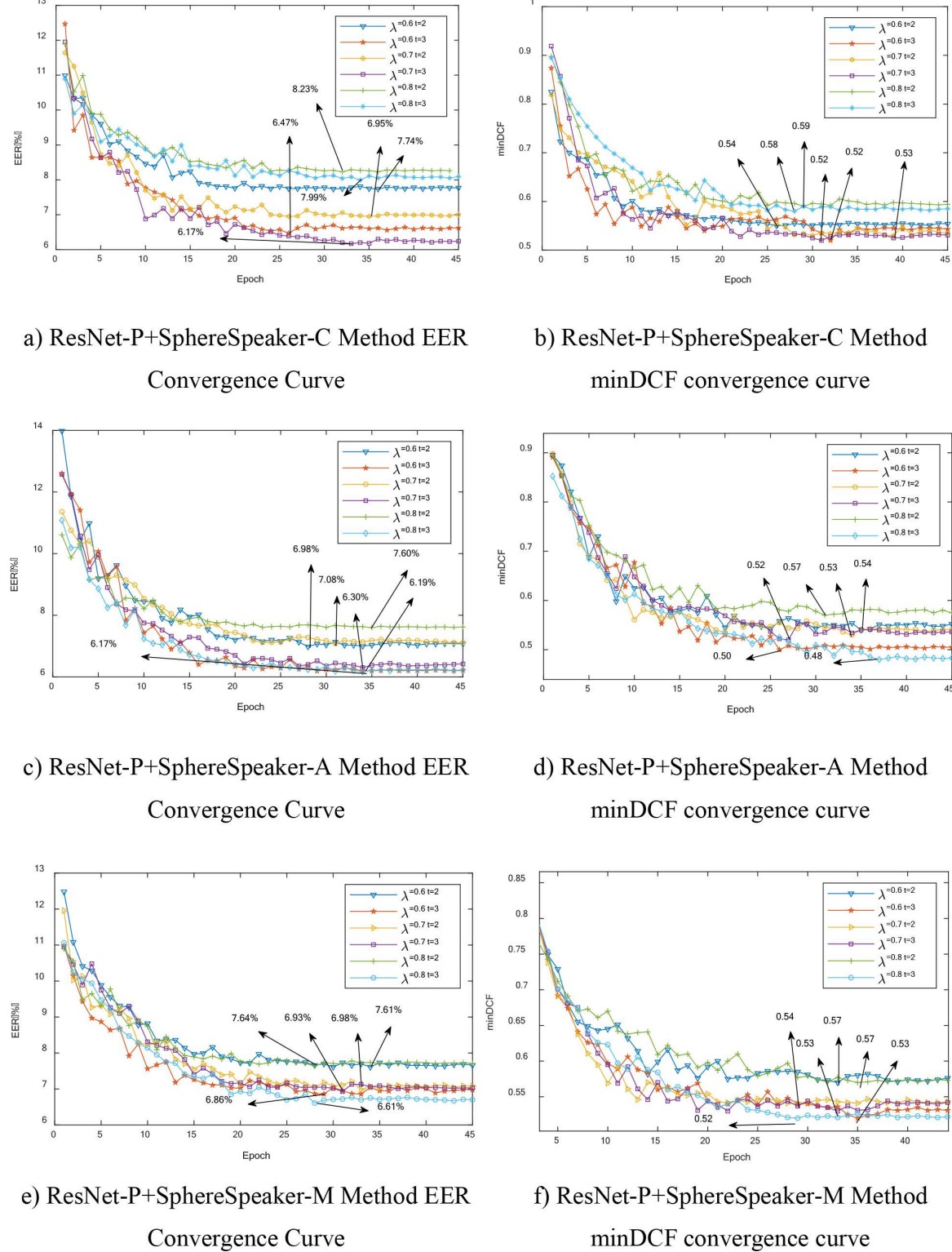

a) ResNet-P+SphereSpeaker-C Method EER Convergence Curve

b) ResNet-P+SphereSpeaker-C Method minDCF convergence curve

c) ResNet-P+SphereSpeaker-A Method EER Convergence Curve

d) ResNet-P+SphereSpeaker-A Method minDCF convergence curve

e) ResNet-P+SphereSpeaker-M Method EER Convergence Curve

f) ResNet-P+SphereSpeaker-M Method minDCF convergence curve

**Fig 3. Comparison Chart of Convergence Curves of the Methods Proposed in this Paper.**

the same method under different parameters, it demonstrates the effectiveness of parameters $\lambda, t$ in addressing the gradient imbalance between positive and negative samples and the overlap of similarity between positive and negative samples during the training process of speaker verification tasks.

## Conclusion

This article proposes a speaker verification method based on target sample mining with modified activation residual networks, the method uses an adaptive objective function under three different angular margins. The method, while preserving the advantages of deep networks in expressing speaker characteristics, addresses the issues of gradient vanishing, gradient explosion, and model degradation caused by increasing the number of neural network layers. Simultaneously introducing adaptive objective functions under three different angular margins to update the network further enhances the stability and generalization ability of the network model. The experimental results indicate that all three proposed methods can enhance the accuracy of speaker verification tasks, improve the stability and accuracy of deep networks in expressing speaker characteristics, and effectively enhance the network's representational ability under the supervision of this objective function. From the performance analysis in terms of performance, parameter analysis, and convergence, the method proposed in this paper performs well in speaker verification tasks.

## Author contributions

**Data curation:** Chen Wei.

**Resources:** Ye Peng.

**Software:** Wang Zhou.

**Supervision:** Zhou Shuhang.

**Writing – original draft:** Ji Chaoqun.

**Writing – review & editing:** Ji Chaoqun.

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
