## [Decision Letter · Decision Letter 0]

3 Dec 2024

PONE-D-24-37227Target sample mining with modified activation residual network for speaker verificationPLOS ONE

Dear Dr. Chaoqun,

Thank you for submitting your manuscript to PLOS ONE. After careful consideration, we feel that it has merit but does not fully meet PLOS ONE’s publication criteria as it currently stands. Therefore, we invite you to submit a revised version of the manuscript that addresses the points raised during the review process.

We look forward to receiving your revised manuscript.

Kind regards,

Qian Zhang, Ph.D

Academic Editor

PLOS ONE

2. Please note that PLOS ONE has spec6ific guidelines on code sharing for submissions in which author-generated code underpins the findings in the manuscript. In these cases, all author-generated code must be made available without restrictions upon publication of the work. Please review our guidelines at https://journals.plos.org/plosone/s/materials-and-software-sharing#loc-sharing-code and ensure that your code is shared in a way that follows best practice and facilitates reproducibility and reuse.

3. We note that your Data Availability Statement is currently as follows: [All relevant data are within the manuscript and its Supporting Information files] Please confirm at this time whether or not your submission contains all raw data required to replicate the results of your study. Authors must share the “minimal data set” for their submission. PLOS defines the minimal data set to consist of the data required to replicate all study findings reported in the article, as well as related metadata and methods (https://journals.plos.org/plosone/s/data-availability#loc-minimal-data-set-definition). For example, authors should submit the following data: - The values behind the means, standard deviations and other measures reported; - The values used to build graphs; - The points extracted from images for analysis. Authors do not need to submit their entire data set if only a portion of the data was used in the reported study. If your submission does not contain these data, please either upload them as Supporting Information files or deposit them to a stable, public repository and provide us with the relevant URLs, DOIs, or accession numbers. For a list of recommended repositories, please see https://journals.plos.org/plosone/s/recommended-repositories. If there are ethical or legal restrictions on sharing a de-identified data set, please explain them in detail (e.g., data contain potentially sensitive information, data are owned by a third-party organization, etc.) and who has imposed them (e.g., an ethics committee). Please also provide contact information for a data access committee, ethics committee, or other institutional body to which data requests may be sent. If data are owned by a third party, please indicate how others may request data access.

Additional Editor Comments:

We note that one or more reviewers has recommended that you cite specific previously published works. As always, we recommend that you please review and evaluate the requested works to determine whether they are relevant and should be cited. It is not a requirement to cite these works. We appreciate your attention to this request.

Reviewers' comments:

Reviewer's Responses to Questions

**Comments to the Author**

1. Is the manuscript technically sound, and do the data support the conclusions?

Reviewer #1: Partly

Reviewer #2: Partly

2. Has the statistical analysis been performed appropriately and rigorously? 

Reviewer #1: No

Reviewer #2: I Don't Know

3. Have the authors made all data underlying the findings in their manuscript fully available?

Reviewer #1: No

Reviewer #2: No

4. Is the manuscript presented in an intelligible fashion and written in standard English?

Reviewer #1: No

Reviewer #2: No

5. Review Comments to the Author

Reviewer #1: This paper introduces a novel neural network architecture termed ResNet-P, which is constructed on the foundation of the Residual Network (ResNet) and incorporates a Parametric Rectified Linear Unit (PReLU) activation function. This modification is designed to enhance the network's capability to represent features and to mitigate issues related to gradient flow. Furthermore, the paper proposes an adaptive target function named SphereSpeaker, which is specifically tailored to optimize sample mining in speaker verification tasks. By incorporating strategies for balancing positive and negative samples and for mining challenging samples, the paper further refines the model's ability to handle imbalanced data during training, thereby improving the model's convergence rate and generalization capabilities. To diminish the overlap in similarity scores between positive and negative sample pairs, the paper introduces a method for adjusting similarities. Experiments conducted on the VoxCeleb1 database substantiate the efficacy of the proposed methods. However, the authors must address the following issues:

1. The paper exhibits subpar writing quality, with several sentences not adhering to standard academic expression. For instance, the phrase "This new network is named the Residual Network-PReLu, ResNet-P." should be rephrased for clarity and formality. Additionally, the use of "This chapter proposes..." in the Conclusion section is somewhat perplexing. The arbitrary naming of numerous acronyms also detracts from the manuscript's scholarly tone. It is recommended that the author revise the entire text for clarity, consistency, and adherence to academic writing standards.

2. The current manuscript appears to be a mere assemblage of technical elements without a coherent logical progression, which may impede the readers' comprehension. The author is advised to restructure the organization of the paper to enhance clarity and facilitate understanding.

3. The author's comparison of algorithms is outdated and does not reflect the current state of the art in speaker verification. It is recommended that the author updates the comparative analysis to include the most advanced algorithms in the field, such as transformer-based methods, to ensure the manuscript's relevance and contribution to the academic discourse.

4. The author claims to have proposed a novel neural network architecture, namely ResNet-P (Residual Network-PReLu). However, it appears that numerous publications have already employed the combination of ResNet and PReLU, as referenced in [1-3]. Consequently, the innovative aspect of this manuscript seems insufficient. It is suggested that the author either provides a more substantial contribution or clearly delineates how their work differs from and improves upon the existing literature.

[1] Peng, S., Huang, H., Chen, W., Zhang, L., & Fang, W. (2020). More trainable inception-ResNet for face recognition. Neurocomputing, 411, 9-19.

[2] Trottier, L., Giguere, P., & Chaib-Draa, B. (2017, December). Parametric exponential linear unit for deep convolutional neural networks. In 2017 16th IEEE international conference on machine learning and applications (ICMLA) (pp. 207-214). IEEE.

[3] Zhao, M., Zhong, S., Fu, X., Tang, B., Dong, S., & Pecht, M. (2020). Deep residual networks with adaptively parametric rectifier linear units for fault diagnosis. IEEE transactions on industrial electronics, 68(3), 2587-2597.

5. The placement of the result figures at the end of the manuscript is not reader-friendly, as it disrupts the flow of information. Additionally, in Figure 3, the subfigures (e) and (f) are not aligned, which affects the visual presentation. It is recommended that the author reorganizes the manuscript to present the results in a more logical sequence as they are discussed in the text and ensures that all visual elements, such as figures, are properly aligned and formatted for clarity and aesthetic consistency.

Reviewer #2: The study proposes a methodology to overcome some limitations in using traditional SoftMax methods. The writer proposed three methods and concluded that they enhance the accuracy of speaker verification tasks.

I think the study is trying to propose solutions to some limitations that can be very useful in the field. However, I think the writer should write more clearly to elevate the level of the paper and make it easier to follow. So, my recommendation is that the writers should edit the writing and add a discussion section then re-submit, if possible.

Here are some comments per section

The Abstract

It took the writer long to set the aim for the study. The abstract has to be concise and talks a bit about the background, the aim of the study, the methods used and the results of the study.

- The writers suggested a method to overcome some limitation with using SoftMax, but didn’t mention anything about the findings at the end of the abstract.

- Please avoid using ‘ novel’ when you propose a method/ an idea.

- I would avoid using ‘has drawbacks’ in the abstract, and instead use ‘some limitations’.

- When you talk about training, mention it is in machine learning. The abstract has to be understood without going back to other parts of the manuscript.

- The second sentence needs paraphrasing, it reads as if the discrepancy between predicted probability and the actual distribution is caused by the fact that SoftMax framework is suitable for multi-class classification problem in verification and training. Is that only SoftMax or are there joint functions that are used to make sure that there are no such discrepancies such as using loss function. Either way, please paraphrase for clarity.

- SoftMax instead of softmax

- As in the abstract we need to mention the methods used, it would be beneficial if you could explain in line or two what is the suggested method and what does it do?// what does it add to the ‘traditional methods’.

- You need to include the findings at the end of the abstract.

- The abstract and the conclusion should align, is it only one suggested method or three suggested methods?

Introduction

- The first part of the introduction reads as if the writer is listing the methods, some connection would help coherence.

- Please connect between paragraphs.

- Each paragraph should have a topic sentence that relates to the previous point, then all the sentences in the paragraph should explain link to other studies and you should critically say how is this point support your study follow PEEL & PEELC methods in writing paragraphs, please.

- In stead of saying ‘this section will…’ try to have some connection even with the use of subheadings.

- Add a Space between ‘.’ And the new sentences, please.

- Use references please when you talk about problems and the issues you raised.

- Avoid starting sentences with ‘and’ please.

- When introducing the data, just start talking about the data straight forward, please.

- You should have a subsection after talking about the data, which might be called ‘procedure’ what you did with the data. This helps the readers follow smoothly.

Experimental analysis

The first sentence here has to align with the objective of the study that you mentioned in the abstract and the conclusion. This has to be clarified in writing. Your proposed model is to use deep residual network and adaptive functions to improve speaker verification systems?, so make sure to emphasize that point.

This chapter would benefit from a discussion section where you present your results, explain them in relation to previous work in the field.

Conclusion

o This section starts by saying that there is a speaker verification method, but later on in the paragraph you say “The experimental results indicate that all three proposed methods can enhance the accuracy of speaker verification tasks”. You need to be precise and clear from the first sentence.

o Please paraphrase this sentence “the stability and accuracy of deep networks in expressing speaker characteristics, and from the perspective of performance analysis in various aspects, the methods exhibit good performance”

6. PLOS authors have the option to publish the peer review history of their article (what does this mean?). If published, this will include your full peer review and any attached files.

Reviewer #1: No

Reviewer #2: No

---

## [Author Response · Author response to Decision Letter 0]

6 Jan 2025

The study proposes a methodology to overcome some limitations in using traditional SoftMax methods. The writer proposed three methods and concluded that they enhance the accuracy of speaker verification tasks.

I think the study is trying to propose solutions to some limitations that can be very useful in the field. However, I think the writer should write more clearly to elevate the level of the paper and make it easier to follow. So, my recommendation is that the writers should edit the writing and add a discussion section then re-submit, if possible.

Here are some comments per section

The Abstract

It took the writer long to set the aim for the study. The abstract has to be concise and talks a bit about the background, the aim of the study, the methods used and the results of the study.

-The writers suggested a method to overcome some limitation with using SoftMax, but didn’t mention anything about the findings at the end of the abstract.

Reply:I have added the findings related content after the summary.

-Please avoid using ‘ novel’ when you propose a method/ an idea.

-I would avoid using ‘has drawbacks’ in the abstract, and instead use ‘some limitations’.

-When you talk about training, mention it is in machine learning. The abstract has to be understood without going back to other parts of the manuscript.

Reply：The above content has been modified as required.

-The second sentence needs paraphrasing, it reads as if the discrepancy between predicted probability and the actual distribution is caused by the fact that SoftMax framework is suitable for multi-class classification problem in verification and training. Is that only SoftMax or are there joint functions that are used to make sure that there are no such discrepancies such as using loss function. Either way, please paraphrase for clarity.

Reply:The second sentence has been paraphrasing.

-SoftMax instead of softmax

Reply:The entire text has been modified.

-As in the abstract we need to mention the methods used, it would be beneficial if you could explain in line or two what is the suggested method and what does it do?// what does it add to the ‘traditional methods’.

Reply: The relevant explanation has been added at the summary.

SphereSpeaker introduces different types of hyperparameters on the basis of Softmax, making it more suitable for handling speaker verification problems.SphereSpeaker also introduces three different angular margins to update the network, further enhancing the stability and generalization ability of the network model.

-You need to include the findings at the end of the abstract.

Reply: Added the discovery at the end of the summary.

The experimental results indicate that compared to other deep neural network methods, this method has the lowest equal error rate, significantly improving the performance of the speaker verification system.

-The abstract and the conclusion should align, is it only one suggested method or three suggested methods?

Reply: Modified the description of the method in the abstract and conclusion. Abstract:SphereSpeaker also introduces three different angular margins to update the network, further enhancing the stability and generalization ability of the network model.

Conclusion:This article proposes a speaker verification method based on target sample mining with modified activation residual networks, the method uses an adaptive objective function under three different angular margins.

Introduction

-The first part of the introduction reads as if the writer is listing the methods, some connection would help coherence.

-Please connect between paragraphs.

Reply: Connections have been added between paragraphs.

-Each paragraph should have a topic sentence that relates to the previous point, then all the sentences in the paragraph should explain link to other studies and you should critically say how is this point support your study follow PEEL & PEELC methods in writing paragraphs, please.

-In stead of saying ‘this section will…’ try to have some connection even with the use of subheadings.

Reply: Modified all sentences containing 'this section will...' in the article.

-Add a Space between ‘.’ And the new sentences, please.

Reply: The relevant content has been modified.

-Use references please when you talk about problems and the issues you raised.

-Avoid starting sentences with ‘and’ please.

Reply: All sentences starting with "and" have been modified.

-When introducing the data, just start talking about the data straight forward, please.

Reply:Content has been deleted, ensuring data is directly discussed.

-You should have a subsection after talking about the data, which might be called ‘procedure’ what you did with the data. This helps the readers follow smoothly.

Reply：The procedure section has been added.

Experimental analysis

The first sentence here has to align with the objective of the study that you mentioned in the abstract and the conclusion. This has to be clarified in writing. Your proposed model is to use deep residual network and adaptive functions to improve speaker verification systems?, so make sure to emphasize that point.

Reply: The method descriptions in the abstract and conclusion have been modified to ensure that all descriptions of the research methods in the article are consistent.This chapter would benefit from a discussion section where you present your results, explain them in relation to previous work in the field.

Conclusion

oThis section starts by saying that there is a speaker verification method, but later on in the paragraph you say “The experimental results indicate that all three proposed methods can enhance the accuracy of speaker verification tasks”. You need to be precise and clear from the first sentence.

Reply: The description of the first sentence has been modified.

“This article proposes a speaker verification method based on target sample mining with modified activation residual networks, the method uses an adaptive objective function under three different angular margins.”

oPlease paraphrase this sentence “the stability and accuracy of deep networks in expressing speaker characteristics, and from the perspective of performance analysis in various aspects, the methods exhibit good performance”

Reply: This sentence has been paraphrase.

“improve the stability and accuracy of deep networks in expressing speaker characteristics, and effectively enhance the network's representational ability under the supervision of this objective function. ”

 Review Comments to the Author

Reviewer #1: This paper introduces a novel neural network architecture termed ResNet-P, which is constructed on the foundation of the Residual Network (ResNet) and incorporates a Parametric Rectified Linear Unit (PReLU) activation function. This modification is designed to enhance the network's capability to represent features and to mitigate issues related to gradient flow. Furthermore, the paper proposes an adaptive target function named SphereSpeaker, which is specifically tailored to optimize sample mining in speaker verification tasks. By incorporating strategies for balancing positive and negative samples and for mining challenging samples, the paper further refines the model's ability to handle imbalanced data during training, thereby improving the model's convergence rate and generalization capabilities. To diminish the overlap in similarity scores between positive and negative sample pairs, the paper introduces a method for adjusting similarities. Experiments conducted on the VoxCeleb1 database substantiate the efficacy of the proposed methods. However, the authors must address the following issues:

1. The paper exhibits subpar writing quality, with several sentences not adhering to standard academic expression. For instance, the phrase "This new network is named the Residual Network-PReLu, ResNet-P." should be rephrased for clarity and formality. Additionally, the use of "This chapter proposes..." in the Conclusion section is somewhat perplexing. The arbitrary naming of numerous acronyms also detracts from the manuscript's scholarly tone. It is recommended that the author revise the entire text for clarity, consistency, and adherence to academic writing standards.

Reply: Modifications have been made to the above question, all sentences containing "This chapter proposes..." have been modified.

2. The current manuscript appears to be a mere assemblage of technical elements without a coherent logical progression, which may impede the readers' comprehension. The author is advised to restructure the organization of the paper to enhance clarity and facilitate understanding.

Reply: Modified part of the paper structure, for example: after the data section, I added a subsection called "Procedure" that explains what I did with the data. This helps the reader's understanding.

3. The author's comparison of algorithms is outdated and does not reflect the current state of the art in speaker verification. It is recommended that the author updates the comparative analysis to include the most advanced algorithms in the field, such as transformer-based methods, to ensure the manuscript's relevance and contribution to the academic discourse.

Reply: In the Performance comparison and analysis section, three new comparison methods from the past two years have been added, and all three methods use the same dataset as this article, voxceleb1.

4. The author claims to have proposed a novel neural network architecture, namely ResNet-P (Residual Network-PReLu). However, it appears that numerous publications have already employed the combination of ResNet and PReLU, as referenced in [1-3]. Consequently, the innovative aspect of this manuscript seems insufficient. It is suggested that the author either provides a more substantial contribution or clearly delineates how their work differs from and improves upon the existing literature.

[1] Peng, S., Huang, H., Chen, W., Zhang, L., & Fang, W. (2020). More trainable inception-ResNet for face recognition. Neurocomputing, 411, 9-19.

[2] Trottier, L., Giguere, P., & Chaib-Draa, B. (2017, December). Parametric exponential linear unit for deep convolutional neural networks. In 2017 16th IEEE international conference on machine learning and applications (ICMLA) (pp. 207-214). IEEE.

[3] Zhao, M., Zhong, S., Fu, X., Tang, B., Dong, S., & Pecht, M. (2020). Deep residual networks with adaptively parametric rectifier linear units for fault diagnosis. IEEE transactions on industrial electronics, 68(3), 2587-2597.

Reply: The method of Resnet+PRelu is indeed applied in other deep learning fields. This paper considers applying the Resnet+PRelu network to the field of speaker verification. In the field of speaker verification, the application of this method is relatively rare. Through experimental comparison, it is found that using the Resnet+PRelu network has relatively good performance. At the same time, the article also made certain modifications to the description of the Resnet+PRelu method, deleting words such as "novel" and "new network". It has been changed to applying Resnet-P to the field of speaker verification.

5. The placement of the result figures at the end of the manuscript is not reader-friendly, as it disrupts the flow of information. Additionally, in Figure 3, the subfigures (e) and (f) are not aligned, which affects the visual presentation. It is recommended that the author reorganizes the manuscript to present the results in a more logical sequence as they are discussed in the text and ensures that all visual elements, such as figures, are properly aligned and formatted for clarity and aesthetic consistency.

Reply: Thank you for the guidance, the figures in the article have been organized to ensure good visual presentation.

---

## [Decision Letter · Decision Letter 1]

16 Feb 2025

Target sample mining with modified activation residual network for speaker verification

PONE-D-24-37227R1

Dear Dr. Chaoqun,

We’re pleased to inform you that your manuscript has been judged scientifically suitable for publication and will be formally accepted for publication once it meets all outstanding technical requirements.

Kind regards,

Qian Zhang, Ph.D

Academic Editor

PLOS ONE

Additional Editor Comments (optional):

Reviewers' comments:

Reviewer's Responses to Questions

**Comments to the Author**

1. If the authors have adequately addressed your comments raised in a previous round of review and you feel that this manuscript is now acceptable for publication, you may indicate that here to bypass the “Comments to the Author” section, enter your conflict of interest statement in the “Confidential to Editor” section, and submit your "Accept" recommendation.

Reviewer #1: All comments have been addressed

Reviewer #2: All comments have been addressed

2. Is the manuscript technically sound, and do the data support the conclusions?

Reviewer #1: Yes

Reviewer #2: Yes

3. Has the statistical analysis been performed appropriately and rigorously? 

Reviewer #1: Yes

Reviewer #2: Yes

4. Have the authors made all data underlying the findings in their manuscript fully available?

Reviewer #1: Yes

Reviewer #2: No

5. Is the manuscript presented in an intelligible fashion and written in standard English?

Reviewer #1: Yes

Reviewer #2: Yes

6. Review Comments to the Author

Reviewer #1: The authors have done a good job of addressing the issues I raised and all comments have been addressed and recommended for publication in this journal.

Reviewer #2: (No Response)

7. PLOS authors have the option to publish the peer review history of their article (what does this mean?). If published, this will include your full peer review and any attached files.

Reviewer #1: No

Reviewer #2: No

---

## [Editor Report · Acceptance letter]

PONE-D-24-37227R1

PLOS ONE

Dear Dr. Chaoqun,

I'm pleased to inform you that your manuscript has been deemed suitable for publication in PLOS ONE. Congratulations! Your manuscript is now being handed over to our production team.

Kind regards,

on behalf of

Dr. Qian Zhang

Academic Editor

PLOS ONE